# TLR3 forms a highly organized cluster when bound to a poly(I:C) RNA ligand

Chan Seok Lim[1], Yoon Ha Jang[1], Ga Young Lee[1], Gu Min Han[1], Hye Jin Jeong[2], Ji Won Kim [1,2] ✉ & Jie-Oh Lee [1,2] ✉

Toll-like Receptor 3 (TLR3) initiates a potent anti-viral immune response by binding to double-stranded RNA ligands. Previous crystallographic studies showed that TLR3 forms a homodimer when bound to a 46-base pair RNA ligand. However, this short RNA fails to initiate a robust immune response. To obtain structural insights into the length dependency of TLR3 ligands, we determine the cryo-electron microscopy structure of full-length TLR3 in a complex with a synthetic RNA ligand with an average length of ~400 base pairs. In the structure, the dimeric TLR3 units are clustered along the double-stranded RNA helix in a highly organized and cooperative fashion with a uniform inter-dimer spacing of 103 angstroms. The intracellular and transmembrane domains are dispensable for the clustering because their deletion does not interfere with the cluster formation. Our structural observation suggests that ligand-induced clustering of TLR3 dimers triggers the ordered assembly of intracellular signaling adaptors and initiates a robust innate immune response.

Toll-like Receptor 3 (TLR3) plays a key role in anti-viral defenses by inducing a potent immune response against double-stranded RNAs (dsRNAs) produced by viral infection[1]. TLR3-deficient mice are susceptible to RNA viruses, such as West Nile virus (WNV)[2], Semliki Forest virus[3], and encephalomyocarditis virus (EMCV)[4] as well as DNA viruses, such as murine cytomegalovirus (MCMV)[5] and herpes simplex virus-1 (HSV-1)[6]. For DNA viruses, TLR3 is proposed to recognize RNA intermediates generated during replication[7]. TLR3 deficiency in humans causes an increase in HSV-1 infection rates[6]. Hyperactive TLR3 responses can have adverse effects, such as virus-induced asthma[8].

Humans have ten TLR family proteins. The binding of their ligands to the extracellular domains causes dimerization of the receptor complexes and triggers the recruitment of common adaptor proteins to the intracellular TIR domains. Five TIR-containing adaptor proteins are known to be involved in the TLR signaling pathway[9]. Among them, TLR3 uses TRIF, also known as TICAM-1, for signal activation[10–12]. Upon binding of dsRNA, TRIF is recruited to the dimerized TLR3, which activates interferon regulatory factor-3 (IRF-3), nuclear factor-kappa B

(NF-κB), and activator protein-1 (AP-1), leading to the induction of type I interferon (IFN), especially IFN-β, cytokine/chemokine production, and dendritic cell maturation[13–15]. TRIF consists of an N-terminal proline-rich domain, a TIR domain, and a C-terminal proline-rich domain. The TIR domain of TRIF is essential for binding to the intracellular TIR domain of TLR3 and also to the signal accessory protein TRAM, also called TICAM-2. All other TLRs except TLR3 use MyD88 for intracellular signaling. MyD88 also contains a TIR domain that is critical for its interaction with receptors and self-oligomerization[16–19]. TLR4 and TLR2 can use both TRIF and MyD88-mediated signaling pathways.

Several synthetic dsRNAs are known to activate TLR3. Among them, polyriboinosinic: polyribocytidylic acid (poly(I:C)) is among the most potent immune inducer of TLR3 activation[20]. It can elicit IFN-α/β production and natural killer (NK) cell activation. It is commonly used to target TLR3 in vaccine adjuvant constructs, and several are under clinical trials[21,22]. Rintatolimod, polyIC$_{12}$U, is a synthetic derivative of poly(I:C) and a highly specific agonist of TLR3[23]. It is currently being used in the clinic to treat chronic fatigue syndrome and is being tested in phase I/II clinical trials against breast, renal cell, and pancreatic

[1]Department of Life Sciences and POSTECH, Pohang 37673, Korea. [2]Institute of Membrane Proteins, POSTECH, Pohang 37673, Korea.
✉e-mail: mokona5@postech.ac.kr; jieoh@postech.ac.kr

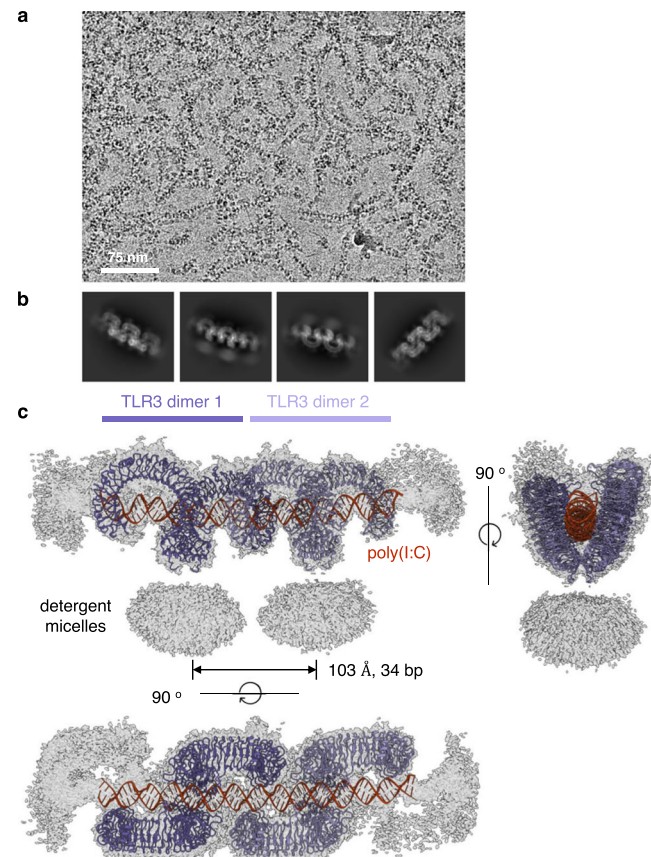

**Fig. 1 | Cryo-EM structure of full-length TLR3 in complex with poly(I:C) RNA.**
**a** Representative micrographic image of the TLR3-poly(I:C) complex. TLR3 forms regularly spaced and linear clusters along the poly(I:C) RNA strands.
**b** Representative 2D class average of the TLR3-poly(I:C) complex. **c** Electron density map of the TLR3-poly(I:C) complex. Atomic models of two neighboring TLR3 dimers are superimposed. The transmembrane and intracellular domains are not visible in the map, presumably due to structure flexibility at the juxtamembrane regions. A representative micrograph from an $n = 9527$ cryo-EM experiment set is shown.

cancer[24]. In addition, an antagonistic antibody against TLR3 is being investigated as an anti-asthmatic agent[25].

Previous crystallographic studies show that RNA ligand binds to the extracellular domains of TLR3 and induces the formation of a homodimer of TLR3[26]. The charge interaction with the phosphate backbone of the RNA double helix plays the principal role in the interaction. The ligand-binding site involves both the N and C-terminal parts of the TLR3 ectodomain. These two interaction sites are separated by ~120 angstrom, which explains why a minimum of 40–50 base pairs (bp) are required for the stable binding of dsRNA and dimerization of TLR3. Receptor dimerization by this short dsRNA molecule is necessary but not enough for a complete immune response because dsRNAs smaller than ~50 bp can activate only the basal activity of TLR3. A robust immune response requires at least ~90 bp of RNA[27–29]. Based on this, it has been proposed that TLR3 forms a cluster on RNA strands, and this ligand-induced clustering initiates the aggregation of intracellular TIR-containing adaptors by binding to the clustered TIR domains of the receptors. However, the structure of this cluster has not been investigated in high resolution.

Here, we report the structure of a TLR3 cluster induced by the binding of a long dsRNA ligand. By mixing full-length TLR3 with poly(I:C) RNA, we form TLR3-dsRNA complexes and determine their structures by cryo-electron microscopy (cryo-EM). The structure shows that TLR3 forms a highly organized and cooperative complex on

the dsRNA helix. Inhibiting the formation of this cluster using an antibody disrupts the full activation of TLR3. The intracellular and transmembrane domains of TLR3 are dispensable for cluster formation.

## Results

### Structure of the TLR3-dsRNA complex

To determine the structure of the TLR3-dsRNA complex, we mixed purified full-length TLR3 with a poly(I:C) dsRNA. The length of the poly(I:C) used in the experiments ranged from 200 bp to 1,500 bp, with an average length of ~400 bp (Supplementary Fig. 1). After a short incubation, the mixture was frozen in EM grids and the EM images were obtained. The raw micrographs showed that a vast majority of TLRs are linearly assembled along the RNA strands with regular spacing (Fig. 1). To determine the high-resolution structure of the TLR3 cluster, the box size for particle picking was adjusted to accommodate two TLR3 dimers. The final reconstituted three-dimensional structure shows a 2.8 Å resolution image of the TLR3 tetramers on the RNA strands (Fig. 1c, Supplementary Table 1 and Supplementary Figs. 2 and 3). The cryo-EM map resolution was estimated to be 2.8 Å, but the RNA region between the TLR3 dimeric units has structural flexibility that prohibited refinement to a higher resolution. However, when the refinement was focused on the dimeric area of the cluster, a 2.3 Å resolution could be achieved. In the structure, the RNA double helix is bent by ~8 degrees at the junction between the two TLR3 dimeric units (Fig. 1c and Supplementary Fig. 4).

To determine if the TLR3 structure is disturbed by linear clustering, we compared our TLR3-poly(I:C) structure to the previously reported crystal structure[26] of the complex between the mouse TLR3 ectodomain and a 46 bp dsRNA. As shown in Supplementary Fig. 5, our cryo-EM structure of TLR3 is practically identical to the reported crystal structure with a Cα root mean square deviation (r.m.s.d.) of 0.77 Å. This result demonstrates that linear clustering does not disturb the structure of the dimeric unit of the TLR3-dsRNA complex. All the TLR3 and RNA backbone interactions found in the previous crystal structure remain identical in our clustered structure; both the N- and C-terminal residues of the TLR3 ectodomain make close contact with the dsRNA and the orientation between the two TLR3s in the dimeric unit remains identical in the cluster (Supplementary Fig. 6). Residues corresponding to His39, His60, His539, and Asn541 of mouse TLR3, which are critical in RNA binding, have identical conformation in the clustered TLR3 structure and interact with the phosphate groups of dsRNA as previously reported.

### Structure of the ectodomain of the TLR3-dsRNA complex and cooperative nature of TLR3 clustering

To determine whether the transmembrane and intracellular domains are necessary for the linear clustering of TLR3, we produced the ectodomain of TLR3 (ectoTLR3) and determined its structure in complex with poly(I:C) dsRNA (Supplementary Table 2 and Supplementary Figs. 7–10). As shown in Fig. 2a, the electron micrographs show that the majority of the ectoTLR3s are in the clustered state. The structure of the ectoTLR3-dsRNA complex is not changed by the deletion mutation and is superimposable with that of the full-length TLR3-dsRNA complex with an r.m.s.d. of 0.42 Å (Fig. 2b, c and Supplementary Fig. 11). Interestingly, the C-terminal ends of the TLR3 ectodomains in the cluster point to the same direction even without the transmembrane and intracellular domains. This observation demonstrates that anchoring to the membrane is not necessary to fix the orientations of the transmembrane and intracellular domains in the cluster.

To study the cooperativity of the clustering, we titrated the ratio of ectoTLR3 molecules against the number of available binding sites in the poly(I:C) (Fig. 3 and Supplementary Table 3). For the ratio estimation, we assumed that the number of binding sites on a single RNA

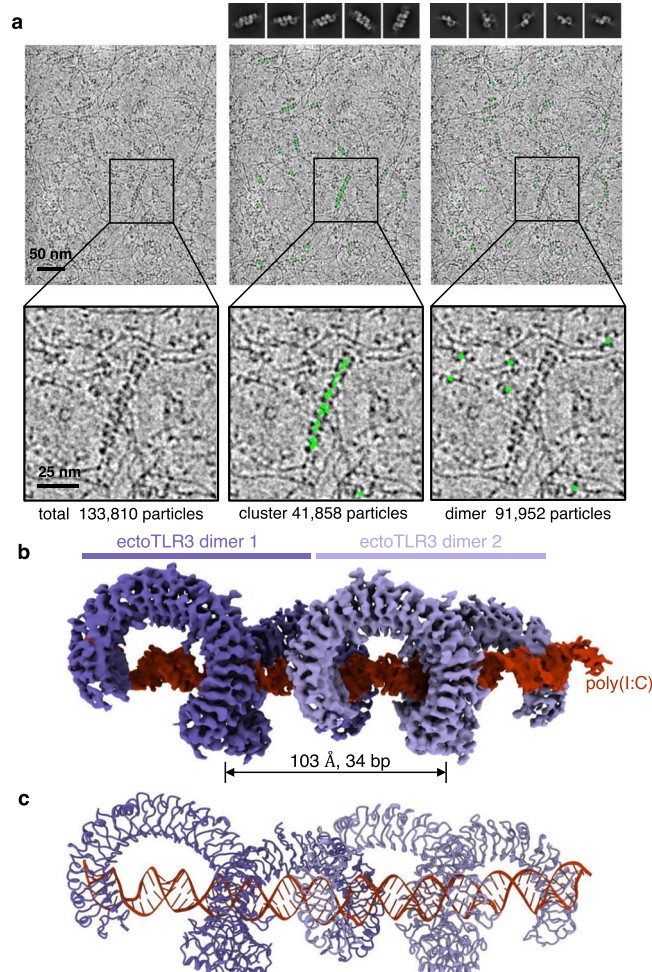

total 133,810 particles    cluster 41,858 particles    dimer 91,952 particles

ectoTLR3 dimer 1    ectoTLR3 dimer 2

103 Å, 34 bp

poly(I:C)

**Fig. 2 | Cryo-EM structure of the ectodomain of TLR3 in complex with poly(I:C) RNA. a** Representative micrographic image of the ectoTLR3-poly(I:C) complex. The picked particles in the cluster (middle) and the dimeric states (right) are marked with green circles. Representative 2D class averages are shown above. **b** Electron density map of the ectoTLR3-poly(I:C) complex. **c** The atomic model of the ectoTLR3-poly(I:C) complex. Representative micrograph is from an $n = 1087$ cryo-EM experiment set.

molecule with a length of 400 bp is 11.8. This is because the spacing between the TLR3 dimers in the cluster state is 34 bp (Fig. 1c). As shown in Fig. 3d, even in the most poly(I:C) excess condition, the electron microscopic images show that a significant portion of the TLR3 particles still is in the clustered state along the dsRNA. In the subsequent 2D classification analysis, the number of the TLR3 particles in the cluster was estimated to be 50% of those in the dimeric state. Furthermore, this ratio increased to 250% when the relative concentration of ectoTLR3 increased (Fig. 3d). In a close inspection of the micrographs, we found that almost all of the particles bound to the straight double-helical part of the dsRNA are in the clustered state (Figs. 1a and 2a). The vast majority of TLR3 not in the clustered state is located either where the TLR3 clusters ended or where the RNA strands make irregular kinks. This observation demonstrates that clustering is a dominant feature for TLR3 when bound to RNAs with regular double-helical structures.

### Disruption of TLR3 clustering using a blocking antibody

To study the importance of cluster formation in TLR3 activation, we determined the structure of the ectoTLR3-dsRNA complex with a bound antibody fragment (Fig. 4, Supplementary Table 4 and

Supplementary Figs. 12 and 13). Previously, Luo et al. showed that mAb12 and mAb1068 antibodies can block the activity of TLR3 without disrupting the binding of dsRNA or dimerization of the receptor. From the crystal structure of the ectoTLR3-antibody complex, they found that the binding sites of mAb12 and mAb1068 antibodies do not overlap with the RNA binding sites. Therefore, they proposed that the antibodies may interfere with TLR3 activation by blocking TLR3 clustering. To confirm this hypothesis, we produced a single-chain variable domain fragment (scFv) form of the mAb12 antibody and tested if it can block TLR3 clustering when bound to poly(I:C) dsRNA. As shown in Fig. 4a, the linear clustering of TLR3 is almost completely prohibited by the antibody even though the molar ratio of antibody vs. TLR3 is 0.5. Those TLR3 particles that still remain in the clustered state do not have the bound antibody in the structure. A similar result was obtained using full-length TLR3 and a Fab form of mAb12 (Supplementary Figs. 14 and 15 and Supplementary Table 5). We also confirmed the previous result, that the mAb12 antibody could inhibit TLR3 activation, in transfected cells, as shown in Fig. 4c, d. This was not due to conformation changes in TLR3 because antibody binding does not disturb the structure or dimerization of TLR3 (Supplementary Fig. 16). Collectively, our structural study and previous reports demonstrate that cluster formation is indispensable for TLR3 signaling.

Close inspection of the region between the TLR3 dimeric units in the cluster shows that the charge interaction plays a role in the clustering (Fig. 5a). The closest contact area between the dimeric TLR3 units is composed of the negatively charged N-terminal and the positively charged C-terminal residues of the neighboring ectoTLR3 dimers. Some of these residues do not directly interact with the RNA (Supplementary Fig. 17). The distance between these charged areas is more than 6 angstroms and appears too far to be considered a strong interaction. However, previous studies showed that even these weak electrostatic forces can modulate protein-protein complex formation[30,31]. To test their role in TLR3 clustering, we made two charge-reversion mutations. The N-terminal mutant, NT-mut had a triple mutation of K117D, K139D and K145D and the C-terminal mutant, CT-mut had a triple mutation of D523K, D524K and E527K of ectoTLR3. For the cryo-EM study, the mutant ectoTLR3 domains were produced and mixed with poly(I:C). The micrographs of the resulting complexes show that the clustering of ectoTLR3 is significantly reduced or entirely blocked by the mutations (Fig. 5b and c, Supplementary Table 4 and Supplementary Figs. 18–21). As shown in Fig. 5d, the dimeric structure of the mutant ectoTLR3 is still superimposable with that of the wild-type ectoTLR3. Therefore, the effect of the mutation is not due to a conformational change of the TLR3 nor the blocking of TLR3 dimerization. To test the importance of the mutated residues in TLR3 signaling for the full-length protein under cellular environment, we transfected the wild-type and a charge-reversed mutant into HEK-Blue cells and tested their responses to the poly(I:C) ligand (Supplementary Fig. 22). The mutant protein is still able to generate an intracellular signal in response to poly(I:C). This result indicates that the charge interaction between the TLR3 dimers contributes to but is not the predominant factor for clustering under the cellular environment where TLR3s are bound to the membrane.

Previously, a point mutation in the BB loop that connects the helix B and strand B of the TIR domain has been shown to interrupt TLR signaling, including that of TLR3. To investigate the role of this mutation in TLR3 clustering, we produced a BB loop mutant, A795H, of TLR3 and determined the cluster state of the mutant[32] (Supplementary Fig. 23). As shown in Supplementary Fig. 24 and Supplementary Table 5, the A795H TLR3 formed the ligand-induced cluster with the same spacing and structure as those of the wild-type TLR3. This structure demonstrates that the BB loop mutation is not directly involved in ligand binding or clustering of TLR3 but in its interaction with intracellular signaling proteins.

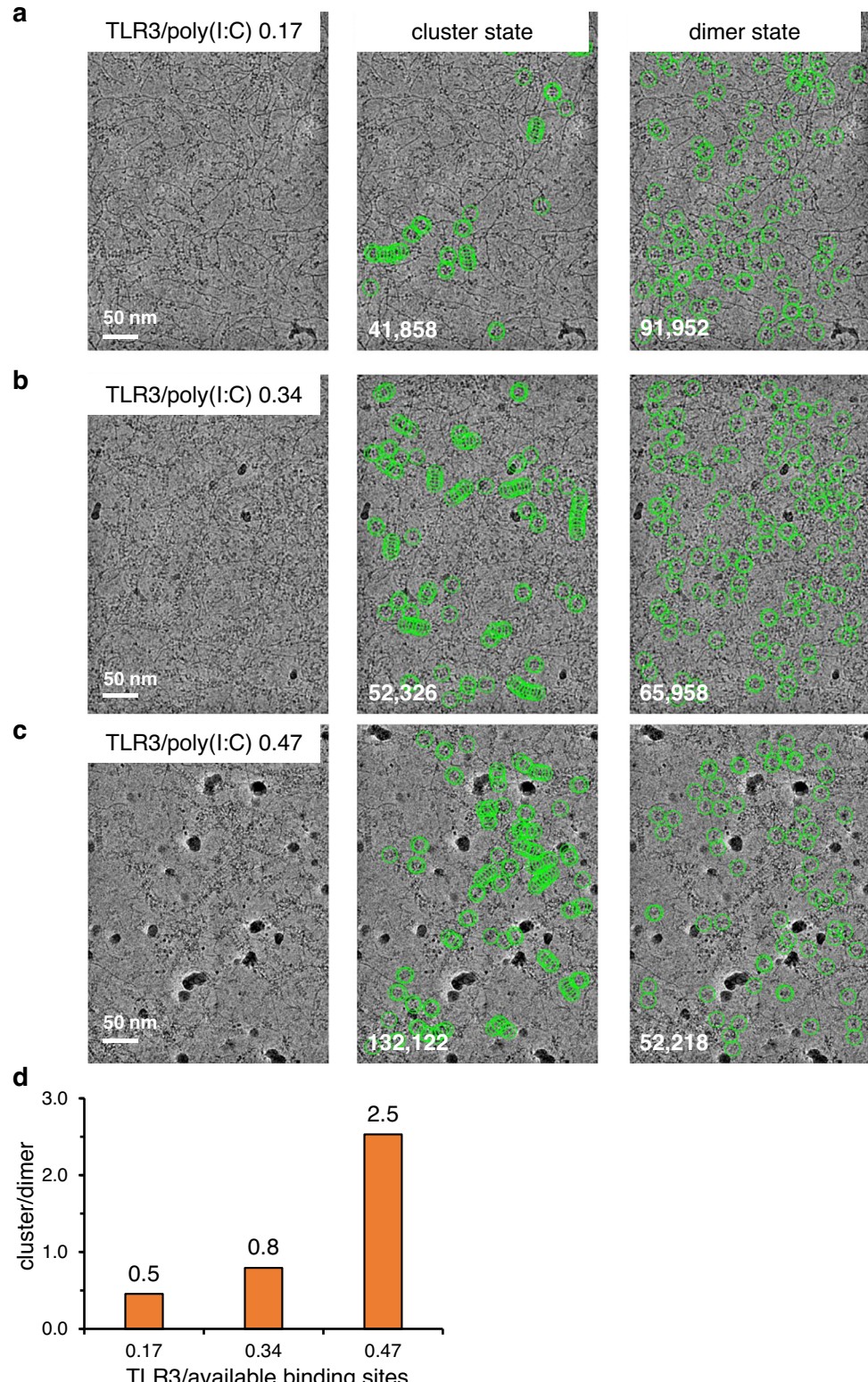

**Fig. 3 | Effect of TLR3 concentration on cluster formation. a–c** Representative micrograph images of the ectoTLR3-poly(I:C) complex with different ratios of TLR3 vs. available binding sites in the poly(I:C). The TLR3s in the clustered (middle) and dimeric states (right) and are marked with green circles. Representative micrographs are from $n = 1087$ (**a**), $n = 1029$ (**b**) and $n = 1081$ (**c**) cryo-EM experiment sets, respectively. **d** The histogram shows the ratio of TLR3 particles in the clustered state vs. the dimeric state by varying the ratio of TLR3 molecules and the TLR3 binding sites available in poly(I:C). Source data are provided as a Source Data file.

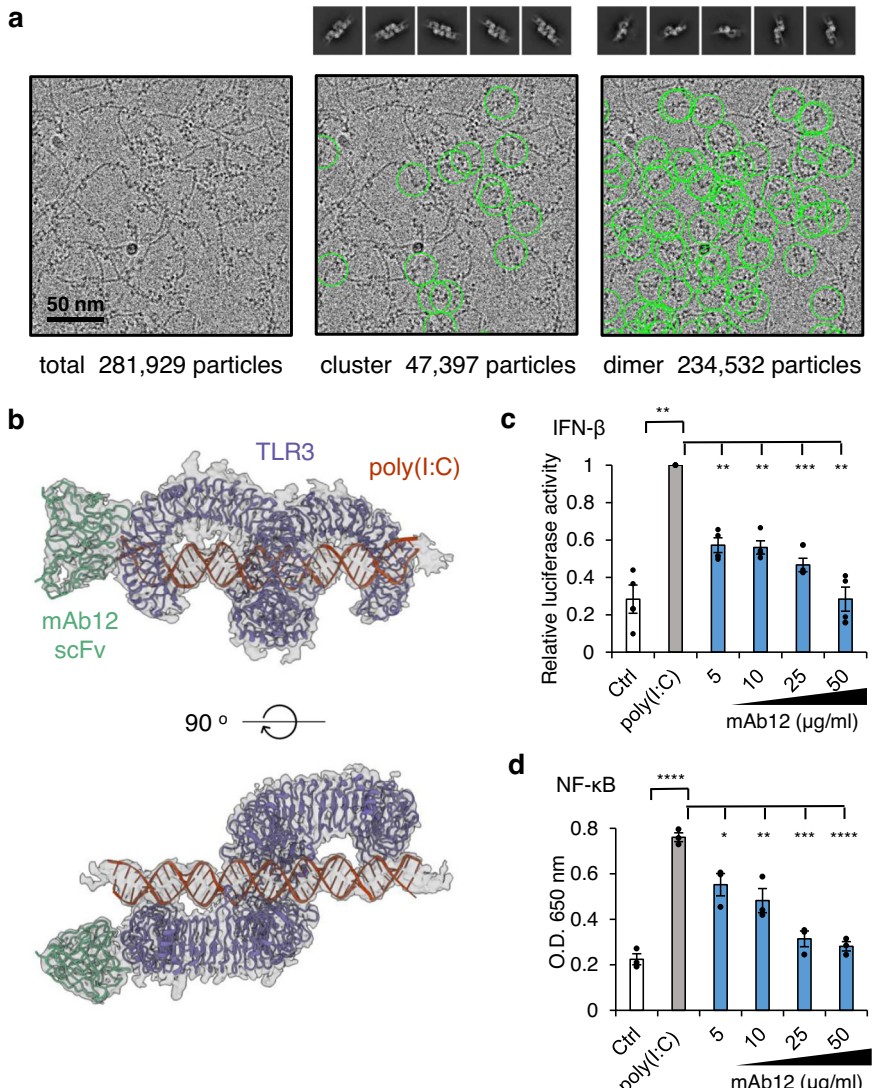

**Fig. 4 | Inhibition of ectoTLR3 clustering by a blocking antibody.**
**a** Representative micrograph (*n* = 1116) of the ectoTLR3-mAb12-poly(I:C) complex.
An scFv form of the antibody mAb12 is used for the complex formation. The TLR3s
in the clustered (middle) and dimeric states (right) are marked with green circles.
The 2D class averaged images of TLR3 are shown above. The molar ratio of mAb12
vs. TLR3 was 0.5:1. **b** Atomic model of the ectoTLR3-mAb12-poly(I:C) complex.
TLR3-dependent IFN-β (**c**) and NF-κB (**d**) assays. TLR3 was stimulated by poly(I:C)

(gray). Reporter signal inhibition by antibody treatment in a dose-dependent
fashion is shown (blue). The cells were pretreated with the antibody one hour
before poly(I:C) treatment. *N* = 4 (**c**) and *n* = 3 (**d**) independent experiments were
performed. Data are presented as mean values ± SEM. Two-tailed *t* test with Welch's
correction (**c**) and two-tailed *t* test without correction (**d**) were used to calculate *p*
values. ns; not significant *; $p < 0.05$, **; $p < 0.01$, ***; $p < 0.001$, ****; $p < 0.0001$.
Source data and *p* values are provided as a Source Data file.

## Discussion

In this research, we found that TLR3 binds cooperatively to a dsRNA
ligand and forms a highly ordered linear cluster. The structures of the
TLR3 subunits in the clustered state remain identical to that of TLR3 in
the dimeric state determined by crystallography. The dimeric units of
TLR3 are repeated 2–20 times with a regular spacing of 103 Å or 34 RNA
base pairs in the cluster. The transmembrane and intracellular domains
are dispensable for cluster formation. The binding of a blocking anti-
body disrupted the cluster formation and immune response without
disturbing RNA binding activity. These results demonstrate that
ordered cluster formation is critical for the full activation of TLR3
against dsRNA ligands.

Previously, Leonard et al. showed that TLR3 had a higher affinity
with longer dsRNAs. They measured the binding affinities of dsRNA
with lengths of 39, 48, 139, and 540 base pairs at three different pHs[27].
They found that, at pH 6.0, the binding affinity of 130 bp dsRNA was
five times higher than that of 48 bp dsRNA. RNA binding showed
strong positive cooperativity, indicating that the affinity increased

with the number of TLRs bound. Similarly, Luo et al. showed that
139 bp dsRNA but not 49 bp dsRNA induced TLR3-dependent NF-κB
activation[29]. They showed that two monoclonal antibodies, mAb12 and
mAb1068, blocked TLR3 activation without disrupting the binding or
dimerization of TLR3. They also showed that mAb12 could block
poly(I:C)-induced CCL5 production in NHBE, a bronchial epithelial cell.
Jelinek et al. showed that dsRNA longer than 90 bp was required to
induce cytokine release by dendritic cells and generate antigen-
specific cytotoxic T cells in mice[28]. These results indicate that clus-
tering is critical for the high-affinity binding and robust activation
of TLR3.

Several factors can contribute to the ordered assembly of TLR3
along the dsRNA strands. First, as shown in Fig. 2, the deletion of both
the transmembrane and the intracellular domains did not change the
structure of the receptor cluster. This result demonstrates that the
ectodomain plays a key role not only in ligand binding and receptor
dimerization but also in receptor clustering. The TLR3-RNA interaction
alone cannot explain the "ordered" clustering because the entropic

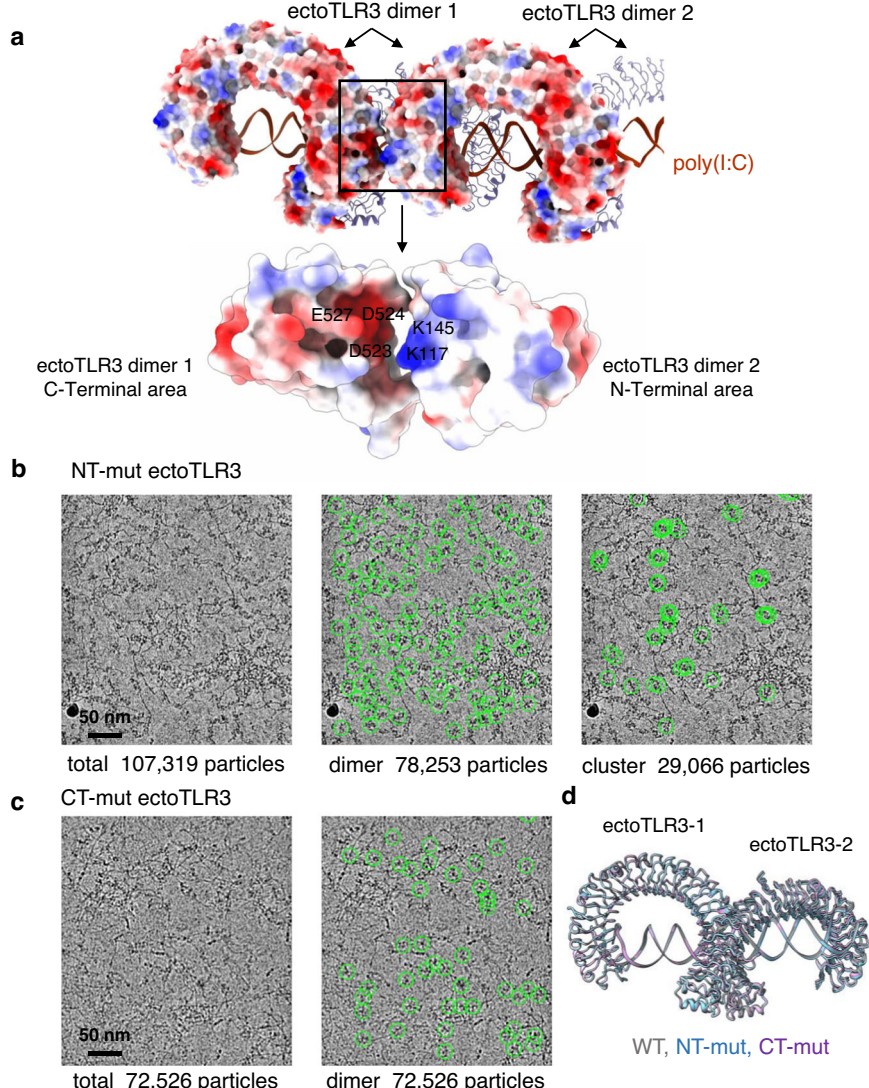

**Fig. 5 | Interaction between TLR3 dimers in the cluster. a** The charge distribution of the TLR3s in the clustered state. The negatively charged and positively charged surfaces are colored in red and blue, respectively. **b** A representative micrograph of the NT-mut mutant (K117D, K139D, K145D). The number of TLR3s in the clustered state is significantly reduced. Representative micrograph is from an $n = 1113$ cryo-EM experiment set. **c** A representative micrograph of the CT-mut mutant (D523K,

D524K, E527K). The mutation practically eliminates the number of TLR3s in the clustered state. Representative micrograph is from an $n = 2010$ cryo-EM experiment set. **d** Structural comparison of wild-type TLR3 against the two charged-reverted mutants. The r.m.s.d. of the NT-mut and CT-mut TLR3s aligned against the wild-type TLR3 is 0.279 Å and 0.147 Å, respectively.

effect will randomly distribute the bound TLR3 dimers along the dsRNA strands. The long-range charge interaction between the ecto-domains of the TLR3 dimers contributes to clustering as shown in Fig. 5. However, it cannot be the predominant factor under the cellular environment where TLR3 are bound to the membrane as shown by the reporter assays (Supplementary Fig. 22). Second, transmembrane and intracellular domains can play indirect roles. The transmembrane and intracellular domains cannot play direct roles in receptor clustering because these domains between the adjacent dimeric units are separated more than 100 angstroms apart. However, they can still play an indirect role. For example, the interaction of the intracellular TIR domain of TLR3 with those of the signaling proteins can stabilize the pre-assembled receptor cluster or amplify the size of the cluster by bringing more receptors to the cluster through TRIF homo-oligomerization. Interactions with other cellular components, such as lipid rafts, intracellular cytoskeletons or proteins such as CLEC18A[33] may also have a role in cluster formation and stabilization. More structural and biochemical research is needed on these effects.

The structure of full-length TLR3 in a complex with UNC93B1 was recently reported by Ishida et al.[34] UNC93B1 is a chaperone protein involved in the trafficking of TLR3 from the endoplasmic reticulum to the endosome. In the structure, UNC93B1 formed a 1:1 complex with TLR3. The main interaction sites are the transmembrane domain and the luminal linker between the transmembrane and the ectodomains of TLR3. Due to this interaction, the ectodomain of TLR3 made a ~30-degree tilt to the membrane plane. The intracellular TIR domain structure is not visible in the cryo-EM map, presumably due to the structural flexibility in the linker between the transmembrane and TIR domains. In this structure, TLR3 is forced to stay inactive by binding to UNC93B1. Therefore, it is unlikely that the transmembrane domain of TLR3 maintains a similar structure when released from UNC93B1. In our structure, the transmembrane domain and the connecting linkers are more flexible and were not resolvable in the electron density map.

Based on our structural observation and previous results, we propose a mechanism of TLR3 activation by dsRNA (Fig. 6). In this

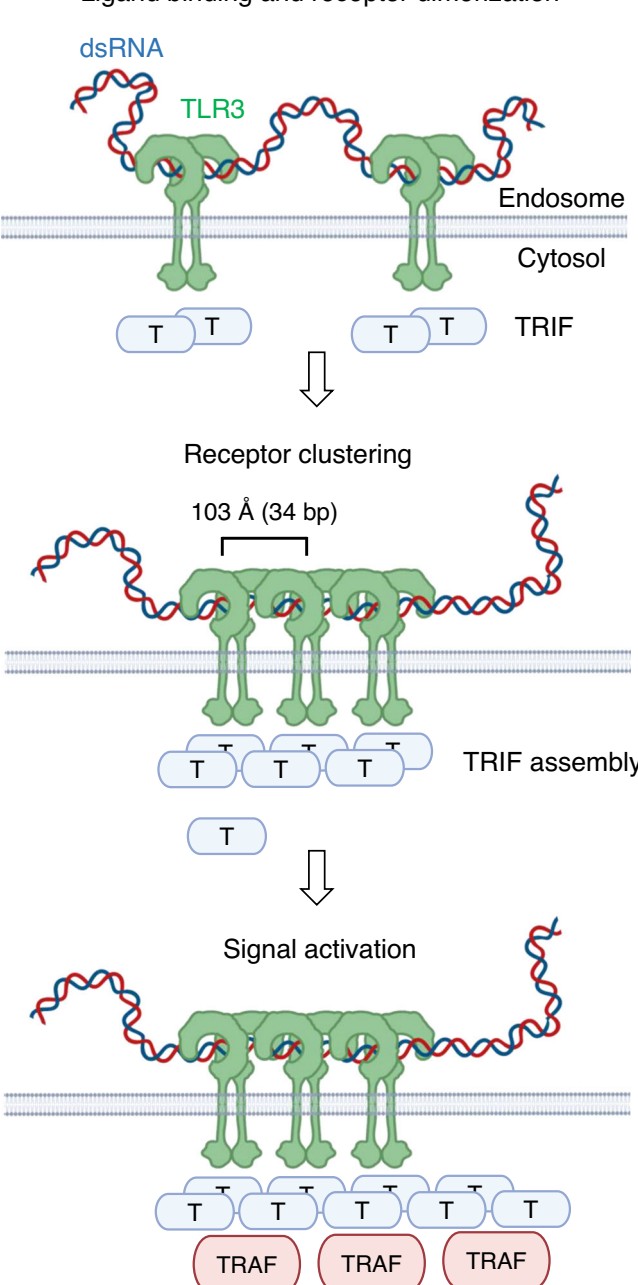

**Fig. 6 | A model of TLR3 activation. TLR3 binds to dsRNA and dimerizes.** The dimeric TLR3s are clustered along the dsRNA strands. Receptor clustering triggers the assembly of TRIF as a linear cluster and recruits downstream effectors.

model, TLR3 binds to the RNA ligand and homo-dimerizes. Then, the interaction between the TLR3 dimers brings the dimeric receptor modules together and assembles a linear cluster of TLR3s. As a result, the intracellular TRIFs start to form a linear assembly. Due to the avidity effect, the clustered TRIFs should have a higher affinity for intracellular signaling proteins, including TRAFs, and initiate an innate immune response. In a previous report, the minimum length of dsRNA that can activate TLR3 was shown to be ~90 bp[27–29] that is sufficient for the binding of two dimeric TLR3 units. This suggests that the formation of the tetrameric assembly of TLR3 is the minimal requirement to trigger the cluster formation of TRIF and initiate downstream signaling. Two factors may contribute to triggering intracellular signaling. First, as mentioned in the previous paragraph, the clustering of the receptors should enhance the binding avidity of TRIF for the receptors.

Second, clustering of the TLR3 TIR domains may increase the productive assembly of intracellular signaling proteins. A previous study showed that TRIF formed a highly ordered and filamentous cluster when activated[35]. The assembly of the TLR3 cluster with proper spacing may enhance the productive formation of the TRIF filament and trigger signaling. Future structural studies on the activated cluster of receptors and signaling proteins are required for a complete understanding of the immune response mediated not only by TLR3 but also by other TLRs.

## Methods

### Expression and purification of full-length TLR3 and the TLR3(A795H) mutant

Synthetic DNA encoding full-length human TLR3 was cloned into pEG BacMam vector[36] (Supplementary Table 6). A 3 C protease site, mCherry, thrombin cleavage site and ALFA tag[37] were added to the C-terminus of TLR3 for detergent screening and purification. All synthetic DNAs were ordered from Twist Bioscience (Supplementary Tables 6-9). The A795H mutation was generated by site-directed mutagenesis using the overlap extension PCR method[38]. The primers used for the mutagenesis experiment are shown in Supplementary Fig. 25. Protein was expressed in HEK293 GnTI⁻ cells cultured in Freestyle media (Thermo Fisher Scientific) supplemented with 1% (v/v) fetal bovine serum (Sigma) in an 8% $CO_2$ shaking incubator. The cells were infected by treating 8% (v/v) baculovirus at a density of $3 \times 10^6$ cells per ml. Expression was enhanced by treating with 10 mM sodium butyrate (Alfa Aesar) after 12 h of infection and further incubated for 48 to 60 h. The cells were collected by centrifugation at 7,261 g for 30 min, flash-frozen in liquid nitrogen, and stored at −80 °C.

For purification, a frozen aliquot of the cells was thawed and resuspended in buffer, 50 mM Tris-HCl pH 8.0, 250 mM NaCl, 20% glycerol, 0.5 mM phenylmethyl fluoride (PMSF), 1 mM benzamidine chloride, 1 µg/ml leupeptin, 1 µg/ml aprotinin, and 1 µg/ml pepstatin and lysed using a microfluidizer (Microfluidics). The membrane fraction was collected by ultracentrifugation, 200,000 g, for one hour. The pellet was resuspended in the lysis buffer and the membrane fraction was solubilized by adding 1% (w/v) Fos-choline 14 (FC-14, Anatrace) and incubated for one hour at 4 °C with gentle rotation. The insoluble fraction was removed by ultracentrifugation at 200,000 g for one hour. The supernatant was loaded onto a column packed with an agarose resin conjugated to an anti-ALFA nanobody[37]. The resin was washed with 40 column volumes of wash buffer containing 20 mM Tris-HCl pH 8.0, 150 mM NaCl, 20% glycerol, and 0.03% FC-14. Protein was eluted by treating with 1% PreScission protease overnight. The eluted solution was concentrated using an ultracentrifugal filter with a 50 kDa cutoff (Merck Millipore). The protein was further purified using a Superose 6 increase 10/300 GL gel filtration column (Cytiva) equilibrated with 20 mM MES buffer pH 5.5, 150 mM NaCl, and 0.03% FC-14. The peak fractions were pooled and concentrated for cryo-EM sample preparation.

### Expression and purification of ectoTLR3 and mutant ectoTLR3

The gene encoding the ectodomain of human TLR3, amino acids 27K-697K, with a thrombin cleavage site and a six-histidine tag at the C-terminus was cloned into a pAcGP67a baculovirus transfer vector (BD Biosciences) (Supplementary Table 7). Recombinant baculovirus was generated by co-transfection with a linearized baculovirus genomic vector, ProGreen™ (AB Vector) into Sf9 insect cells. Protein was expressed in High Five insect cells by adding 4% (v/v) baculovirus at 21 °C for 72–84 h. The secreted protein was bound to cOmplete His-Tag Purification Resin (Roche) and eluted using an elution buffer containing 20 mM Tris-HCl, 150 mM NaCl, and 300 mM imidazole at pH 7.0. The protein was further purified by a Superdex 200 increase 10/300 GL gel filtration column (Cytiva) equilibrated with a buffer containing 20 mM MES pH 5.5, and 150 mM NaCl.

## Expression and purification of the mAb12 antibody

The scFv fragment of the mAb12 antibody was designed by combining the previously reported anti-TLR3 mAb12 variable region sequence (PDB ID: 3ULU) with a flexible linker GGGGSGGGGSGGGGS (Supplementary Table 8). The resulting scFv gene was cloned into the pAcGP67a vector and the recombinant virus was generated with Pro-Green™. Protein was expressed from High Five insect cells by infecting the recombinant baculovirus at 21 °C for 72 h. The secreted protein was purified using cOmplete His-Tag Purification Resin and eluted using an elution buffer containing 20 mM Tris-HCl, 150 mM NaCl, and 300 mM imidazole, pH 7.0. The protein was further purified by a Superdex 200 increase 10/300 GL gel filtration column equilibrated with a buffer containing 20 mM MES, pH 5.5, and 150 mM NaCl.

The heavy and light chain genes for the Fab form of mAb12 were cloned into pEG BacMam vector[36]. A 3 C protease site, mCherry, thrombin cleavage site and ALFA tag[37] sequences were added to the C-terminus of the heavy chain (Supplementary Table 8). A 3 C protease site, eGFP and His$_{10}$-tag sequences were added to the C-terminus of the light chain. The Fab protein was expressed in HEK293 GnTI$^-$ cells by infecting the recombinant heavy and light chain baculoviruses simultaneously. The secreted protein was purified using an agarose resin conjugated to an anti-ALFA nanobody, washed with buffer containing 20 mM Tris-HCl pH 8.0, 150 mM NaCl and eluted using by treating with 1% PreScission protease overnight. The cleaved protein was purified using a Superdex 200 increase 10/300 GL gel filtration column equilibrated with a buffer containing 20 mM MES, pH 5.5, and 150 mM NaCl.

## Cryo-EM sample preparation

Both ectoTLR3-poly(I:C) and the full-length TLR3-poly(I:C) complexes were formed by mixing TLR3 and poly(I:C) (Sigma) and incubated at 4 °C for 30 min. The ectoTLR3 and full-length TLR3-mAb12-poly(I:C) complexes were formed by mixing, in order, antibody, TLR3 and poly(I:C). The mixing ratios of the TLR3-poly(I:C) complexes are summarized in Supplementary Tables 1–5. The 300-mesh Quanti-foil R 1.2/1.3 holy carbon grids (Quantifoil Micro Tools) were discharged at 15 mA for 60 s using a glow-discharger (PELCO). Cryo-EM samples were prepared using a Vitrobot Mark IV (Thermo Fisher Scientific) under 4 °C and 100% humidity condition. Three and a half microliters of the sample were loaded onto the cryo-EM grid and blotted 5–6 s before plunge-freezing.

## Cryo-EM data collection

The data collection statistics are summarized in Supplementary Tables 1–5. Cryo-EM data collection of the full-length TLR3-poly(I:C) and TLR3(A795H)-poly(I:C) complexes were accomplished using a Titan Krios microscope (Thermo Fisher Scientific) at 300 kV and equipped with an energy filter and K3 direct electron detector (Gatan) operating in a counting mode at 100,500 magnification and pixel size of 0.850 Å. Each cryo-EM movie was recorded onto 50 frames with a total dose of 50 e/Å². Automated data acquisition was performed using EPU software for Single Particle Analysis (Thermo Fisher Scientific).

The wild-type and mutant ectoTLR3-poly(I:C) data sets and the full-length TLR3-mAb12-poly(I:C) data sets were acquired using a Talos Arctica microscope (Thermo Fisher Scientific) at 200 kV and equipped with a K3 direct electron detector (Gatan) operating in the counting mode at 100,000x and a pixel size of 0.831 Å. Each movie was recorded onto 50 frames with a total dose of 50 e/Å². Automated data acquisition was performed with EPU for Single Particle Analysis with varying defocus ranges from −0.9 to −2.5 μm.

The ectoTLR3-mAb12-poly(I:C) data set was acquired using a Talos Glacios microscope (Thermo Fisher Scientific) at 200 kV and equipped with a FalconIV direct electron detector (Thermo Fisher Scientific) operating in the counting mode at 92,000x (pixel size 1.113 Å). Each movie was recorded onto 40 frames with a total dose of 40 e/Å².

Automated data acquisition was performed with EPU for Single Particle Analysis with varying defocus ranges from −0.9 to −2.5 μm.

## Cryo-EM data processing and model building

All cryo-EM data were analyzed using the cryoSPARC v3.2 program[39]. The cryo-EM movie files were preprocessed using patch motion correction and the patch contrast transfer function (CTF) methods with parameters estimated by patch CTF estimation method. Poor quality micrographs were rejected based on the CTF estimations, ice thickness, and the total motion of the frames. TLR3 particles in the cluster and dimer states were picked using the Topaz particle-picking method[40]. For this, the TLR3 particles were initially picked by the blob-picking method using the cryoSPARC program. These particles were used to generate the picking models for Topaz. The box sizes for particle picking were adjusted to 360-512 angstroms, which is large enough to accommodate a TLR3 tetramer (Supplementary Tables 1–5). Multiple Topaz iterations were conducted to optimize the accuracy of the initial picking models (Supplementary Fig. 26).

The final picked particles were separated into 50 classes by 2D classification using the cryoSPARC program. Classes of particles with clear and well-centered images were purified by multiple rounds of 2D classifications and refined by the heterogeneous refinement method with multi-class Ab-initio models. The TLR3 particles could be categorized into two different states, clustered and dimeric states, by the 2D classification. The accuracy of the state assignment was confirmed by overlaying the coordinates of the picked particles over the representative microscopic images and manually inspecting the clustering states of the particles (Figs. 2–5). After particle sorting, either the homogeneous refinement or non-uniform refinement method[41] was conducted, and a global refinement map was obtained. The cryo-EM map for dimeric TLR3 was improved by imposing C2 symmetry and a local refinement was conducted to minimize the effect of structural flexibility at the boundaries between the dimeric TLR3 units. Local refinement was conducted by generating masks encompassing the TLR3 dimer. The resulting local refinement maps were combined using the Phenix program[42] to obtain the consensus map of the TLR3 tetramer in the clustered state.

All map resolutions were estimated using a half-maps Fourier shell correlation (FSC) curve by applying a 0.143 threshold. The local resolution estimation and sharpening of the map were conducted using the cryoSPARC program[39,41]. For model building, crystallographic models of a dimeric TLR3 (PDB ID:2A0Z)[43], a mAb12 antibody fragment (PDB ID: 3ULS), and a dsRNA (PDB ID: 3CIY)[26] were used as the initial templates. The initial models were docked into the cryo-EM density map using the UCSF Chimera and the ChimeraX programs[44,45]. An atomic model of the 80 bp dsRNA in the cluster was generated by linking two 46 bp dsRNA models and refined with base-pair and stacking restrains. The resulting models of the TLR3-poly(I:C) complexes were refined through multiple rounds of using Phenix[42] and Coot[46]. The refinement statistics are summarized in Supplementary Tables 1–4.

## Reporter assays and immunoblotting

For the luciferase IFN-β assay, HEK-Blue Null cells (InvivoGen) were seeded at 8×10⁴ cells/well in a 24-well plate and co-transfected with 200 ng of TLR3-pcDNA3.1 plasmid, 50 ng of a reporter plasmid containing a firefly luciferase gene under the control of the IFN-β promoter (IFN-Beta_pGL3, Addgene, Plasmid #102597), and 2 ng of Renilla luciferase gene under the control of a constitutive HSV-thymidine kinase promoter (pRL-TK, Promega) using Lipofectamine 2000 according to the manufacturer's protocol (Supplementary Fig. 27). Transfection vectors containing the N-terminal FLAG tag and the human full-length TLR3 genes were cloned into pcDNA 3.1(+) to generate the TLR3-pcDNA3.1 plasmid (Supplementary Table 9). After 24 h of transfection, the cells were treated with 10 μg/mL of poly(I:C) (InvivoGen) to stimulate TLR3 signaling. To test the neutralizing effect of the mAb12

antibody, the cells were pretreated for one hour with 5 – 50 µg/mL of the scFv form of the antibody before poly(I:C) addition. After 24 h of TLR3 stimulation, the cells were harvested and firefly luciferase and renilla luciferase activity was measured using the Dual-Glo® Luciferase Assay System (Promega) according to the manufacturer's protocol. The samples were transferred to a 96-well plate and luminescence signals were measured using a microplate reader (BioTek). Firefly luciferase activity was normalized to Renilla luciferase activity. All data are presented as the mean ± SEM (error bars). GraphPad Prism 9 (GraphPad Software, USA) was used to perform a $t$ test to identify significant differences between groups, as indicated in the figure legends.

For the Quanti-Blue NF-κB assay, HEK-Blue Null cells, which express a secreted embryonic alkaline phosphatase (SEAP) reporter gene under the control of an IFN-β minimal promoter fused to five NF-κB and AP-1 binding sites, were cultured in Dulbecco's modified Eagle's medium (DMEM; WELGENE) with 5 mM L-glutamate and 1% antibiotic-antimycotic and Zeocin (Thermo Fisher Scientific). The HEK-Blue Null cells were seeded at $8 \times 10^4$ cells/well in a 24-well plate and transiently transfected with the TLR3-pcDNA3.1 plasmid at 70-80% confluency. 500 ng of plasmid DNA and Lipofectamine 2000 (Thermo Fisher Scientific) were used for transfection according to the manufacturer's protocol. After 48 h of transfection, the cells were treated with 10 µg/ml poly(I:C) (InvivoGen) to stimulate TLR3 signaling. To test the neutralizing effect of the mAb12 antibody, the cells were pretreated one hour with 5–50 µg/ml of the scFv form of the antibody before poly(I:C) addition. After 24 h of TLR3 stimulation, the supernatants and cells were collected to quantify NF-κB signaling by a SEAP assay using a Quanti-Blue NF-κB assay kit (InvivoGen). For the assay, 20 µl of the supernatant was reacted with 180 µl of Qunati-Blue reagent in a 96-well plate for 1-3 h at 37 °C, and the absorbance was measured at 650 nm using a microplate reader (BioTeK).

The expression level of TLR3 was measured by immunoblotting using an anti-FLAG-tag antibody. The cell pellets were lysed in RIPA lysis buffer (Merck Millipore) containing a protease inhibitor cocktail (Roche). The samples containing 10 µg of proteins were subjected to 10% sodium dodecyl sulfate-polyacrylamide gel electrophoresis (SDS-PAGE) and transferred onto polyvinylidene fluoride (PVDF) membranes (GE Healthcare). The membranes were blocked with 10% skim milk solution overnight at 4 °C. The membranes were washed with TBS-T buffer containing Tris-buffered saline and 0.1% tween-20 and incubated with an anti-FLAG antibody conjugated with horseradish peroxidase (MBL, cat. no. M185-7, 1:10,000 dilution) for one hour at room temperature. The excess antibody was washed with TBS-T buffer and the membranes were stained with an Immobilon Western kit (Merk Millipore). Images of the immunoblotted gels were recorded using ChemiDoc XRS + (Biorad). Actin was stained using an anti-β-actin antibody (Santa Cruz, cat. no. sc-47778, 1:2,000 dilution) to determine the relative amount of TLR3 in each batch.

### Reporting summary
Further information on research design is available in the Nature Portfolio Reporting Summary linked to this article.

## Data availability
The experiment data that support the findings of this study are available from the corresponding author upon reasonable request. Source data are provided with this paper as a Source Data file. The cryo-EM maps and the atomic coordinates were deposited in the Electron Microscopy Data Bank and the Protein Data Bank under the following accession codes, which are summarized in Supplementary Tables 1–5: EMD-32844, EMD-32845, EMD-32846, EMD-32852, EMD-32853, EMD-32851, EMD-34361, EMD-34367, 7WV3, 7WV4, 7WV5, 7WVF, 7WVJ, 7WVE. Source data are provided with this paper.

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

## Acknowledgements
This work was funded by the National Research Foundation of Korea (2019M3E5D6066058 and 2017M3A9F6029753) and by the Technology Innovation Program, MOTIE of Korea (20019707) to J.-O.L.

## Author contributions
C.S.L., Y.H.J., G.Y.L., G.M.H. performed cloning, mutagenesis, protein purification, cryo-EM experiments, structure determination and refinement, and reporter assays. H.J.J. performed reporter assays. J.W.K. and J.-O.L. supervised the project. The manuscript was written by C.S.L., J.W.K. and J.-O.L. with input from other authors.

## Competing interests
The authors declare no competing interests.
