## [Peer Review File · Nature Communications]

TLR3 forms a highly organized cluster when bound to a poly(I:C) RNA ligandREVIEWER COMMENTS

Reviewer #1 (Remarks to the Author):

In the article “TLR3 forms a highly organized cluster when bound to a poly(I:C) RNA ligand” the authors report cryoEM structures of both full-length TLR3 and a construct only encompassing the ectodomain in complex with long (~400-1500 bp) double stranded RNA fragments. The cryoEM data reveal that the ectodomain of TLR3 form linear assemblies on double stranded RNA and high resolution reconstructions are reported for a TLR3 ectodomain tetramer which consist of two dimers associated head-to-tail. The transmembrane regions and intracellular TIR domains are unfortunately not resolved in the full-length reconstruction. Each ectodomain dimer in the tetrameric complex is almost identical to a previously reported crystal structure of TLR3. Moreover, the tetramers are almost identical in the full-length and ectodomain only reconstructions, demonstrating that the transmembrane regions and the TIR domains does not impact how TLR3 assemble on double-stranded RNA. The two TLR3 ectodomain dimers are in close proximity to each other and have electrostatic complementarity. Incubating an antibody (which previously was shown by crystallography to bind to one of the tetrameric head to tail surface regions reported in this study) with the TLR3 ectodomain leads to reduced clustering based on a cryoEM 2D class analysis. The authors also confirm that this antibody reduces TLR3 signaling activity in a cell-based assay – consistent with already published data. Reverse charge mutations in the head-to-tail interface also lead to reduced clustering based on cryoEM 2D class analysis. However, the mutant is still able to signal in a cell-based assay and the authors speculate that stabilizing interactions involving the transmembrane regions or TIR domains compensate for the effects of reverse charge mutations in the ectodomain head-to-tail-interface.

The concept of TLR3 clustering on double-stranded RNA being required for a robust immune response is well-established. The structural analyses presented in this paper are rigorous and of high quality, and provide important new insight into the molecular basis of how TLR3 cluster on double-stranded RNA. The paper will be of interest to those working in the innate immunity field and molecular mechanisms of signal transduction.

Point that needs to be addressed:

Since the authors have a cryoEM system setup for full-length TLR3 I recommend (if possible) they also test the effect of antibody and reverse-charge mutants on clustering of full-length TLR3 on double-stranded RNA. This should address whether or not the transmembrane or TIR domain regions also contributes to clustering as suggested by the authors.

Many of the conclusions in this study are based on comparing number of particles in clustered and dimeric states from 2D classes analyses of cryoEM data but there is limited method details. Additional details should be provided on the 2D templates used for picking clustered and dimeric particles and how the 2D classification (including numbers of 2D classification rounds) was performed for each dataset.

The authors propose that the linear clustering of ectodomains observed for TLR3 in this study may be a mechanism used by all TLRs (lines 51-53 and 180-191) since they all have intracellular TIR domains. However, only TIR domains of adaptor proteins involved in TLR signalling have been shown to form higher-order assemblies. Such assemblies is yet to be shown for any TLR TIR domains and it is not clear whether or not they need to form larger complexes to induce a robust immune response. Moreover, there is no evidence that other TLR ectodomains cluster upon ligand binding and this seems unlikely for some TLRs that interact with smaller ligands. I recommend the authors moderate/revise these statements.

Extended data figure 6a-c: Number of particles in total and in the clustered and dimeric states based on the 2D class analyses should be included.

Extended data figures 10-15: These figures should be referenced in main text when describing the structures. These figures should also include sub-panels showing fit of model to density and map-vs-model fsc curves.

Minor issues:

Lines 63-64. Two values for the final resolution is reported (2.7 or 2.8 Angstrom). Please update

Extended data table 3: this table is lacking titles for the three different columns.

Extended data figure 4. In the RNA molecule the colours are too similar. Please update sugar/base colour.

Reviewer #2 (Remarks to the Author):

TLR3 is an important pattern recognition factor for dsRNA. Upon detection of viral dsRNAs, TLR3 recruit TICAM/TRIF, causing activation of NF-kappa-B, nuclear translocation of IRF3, cytokine release and anti-viral responses. The signaling unit is composed of one ds-RNA of around 40 bp and two TLR3 molecules. Lateral clustering of signaling units along the length of the ds-RNA ligand is required for TLR3 signal transduction. However, the structure of this cluster has not been investigated in detail. The author reported the cluster of TLR3 along a long dsRNA ligand, which provides novel insight for understanding of TLR3 activation and signal transduction.

This paper is generally well written and clear. I think that the topic is of wide interest to the readership. Nonetheless, if further experiments can be carried out, it would greatly strengthen the paper. More specifically:

1. The displayed structure showed only the ectodomain, without the transmembrane domain and TIR domain. There is structure of full length TLR3 (Ishida H., NSMB, 2021). Maybe the author could consider docking the published structure to acquire a full length TLR3 cluster to observe the interactions between transmembrane domains and TIR domains.

2. The author claimed that transmembrane and TIR domains were dispensable for the clustering, however compared from Fig 1a and Fig 2a, it is obvious that full length TLR3 formed much more clusters than did the ectodomain only proteins. So what are the exact functions and interactions of transmembrane domain and TIR domain in clustering?

3. It is showed that the TLR3 dimer structure is quite stable and easy to form. The C-terminal of ectodomain TLR3 and N-terminal of ectodomain interaction account for clustering. Despite TLR3:dsRNA interaction, does TLR3 interact with TLR3 on the opposite side in the dimer unit?

4. The author claimed that cluster boost stronger immune responses, all they did were in vitro experiments. Can the clustering be observed in cells or in physiological conditions?

5. It seemed the dimer and cluster conformation had no obvious differences, thus how did the protein cluster induce stronger signal activation? Perhaps more downstream molecules?

Reviewer #3 (Remarks to the Author):

Lim C.S. et al. investigated the structure of TLR3 by cryo-EM using a synthetic double-stranded RNA analogue, poly (I:C), with an average length of 400 base pairs and showed that TLR3 dimers form clusters at regular intervals along the RNA ligand. The authors further showed that the intracellular and transmembrane domains of TLR3 are dispensable for cluster formation, as observed using these domains-depleted-recombinant proteins. The authors also demonstrated that these cluster formations are important for TLR3 function using the NF- κ B assay. The authors used scFv form of mAb12, which is previously reported to inhibit TLR3-signaling and clearly showed this antibody blocked the cluster formation of TLR dimers. This data strongly suggests that TLR3 cluster formation is important for full activity sufficient to cause an immune response, whereas the TLR3 dimer is the minimum unit required for the function of TLR3 signaling.

Overall, this study is novel and deeply significant in advancing our knowledge in the mechanism of TLR3-mediated dsRNA recognition. Furthermore, the manuscript is well written. However, there are concerns particularly in the importance of TLR3 inter-dimer cluster formation in the activation of signaling pathways. Whereas the disruption of TLR3 clustering by treating with a blocking antibody nicely inhibited poly (I:C)-induced NF- κ B activation, the C-terminal mutants did not affect the TLR3-signaling despite it inhibited the cluster formation.

Further specific comments to be addressed are as follows.

Major comments:

1) The authors can identify a minimum unit of a TLR3 cluster for triggering the activation of signaling pathways. Is a tetramer formation sufficient, or is the multimer formation (more than hexamers) required? The author can test this by using dsRNAs with different lengths.

2) An analogue of poly (I:C), poly I:poly C12U, has been recognized as a weak TLR3 ligand. Since the lengths of dsRNA part in poly I:poly C12U is supposed to be shorter than poly (I:C) due to a mismatch, it is interesting to investigate if the TLR3 cluster formation is restricted on this compound.

3) The TLR3 signaling leads to the activation of IRF3, which transactivates type I IFN genes and ISGs, in addition to NF- κ B. Thus, the authors need to check if the activation of IFN β promoter is impaired by the inhibition of cluster formation via the antibody treatment and/or mutations. Indeed, the authors described the use of a reporter controlled by the IFN- β promoter in the methods section, although I could not find the data using this reporter in the figures or the main text.

4) The authors found that the TLR3 cluster formation is inhibited by mutations in the N-terminal (NT) part as well as in the C-terminal (CT) part. However, the TLR3 CT mutant had no effect on NF- κ B assay as shown in Extended Fig. 9. The authors need to explain the discrepancy between the antibody treatment and the introduction of mutations in the inter-dimer interface. First, the authors should test the effect of NT mutations in NF- κ B activation. Further, the authors can test if these mutations affect the expression of IFN β -reporter.

5) Extended Data Fig. 6d shows that the ratio of TLR3 cluster to dimeric state greatly elevated when the concentration of ecto TLR3 increased. The results suggest that there is a threshold in the concentration of TLR3 preferentially forming clusters rather than dimers. Since the authors found that the cluster formation contributes to efficient signaling, it is intriguing to examine if the gradual increase of TLR3 expression in cells leads to the great enhancement of signaling at the certain TLR3 expression level.

Minor comments:

- 1) TLR3 should be referred to as Toll-like receptor 3 at the first appearance.
- 2) "r.m.s.d." in line 73 should be spelled out at the first appearance that it stands for root mean square deviation.
- 3) The scFv on line 119 should be spelled out at first appearance as a single-chain variable domain fragments.
- 4) The statement on line 147 that "transmembrane sites and intracellular domains are important for cluster formation." is confusing because it is contradictory to the results shown in Fig. 2. In this figure, the authors show that TLR3 lacking these domains still forms clusters. I believe that the clarification of this sentence is required.

Reviewer #4 (Remarks to the Author):

Lim et al. describe the cryo-EM structures of human TLR3 with poly(I:C) dsRNA. The study determined that TLR3 dimers form a highly ordered and cooperative complex on long dsRNA. Further, the authors solved cryo-EM TLR3:dsRNA structure with the antibody that disrupts the activation of TLR3s. Based on their results, they present the activation model of TLR3 upon binding to dsRNA ligands.

This study is a follow up of the earlier published results (Liu, L. et al. Science. 2008) with more potent and longer dsRNA substrate that causes clustering of TLR3s along the long dsRNA. This work is particularly important to understand the detailed mechanisms of TLR3 activation by dsRNAs. I am very impressed with their cryo-EM structures and the study design in general. The work is complete in my opinion with minor concerns that I have as follows:

1. Did author analyze the dimer1-dimer2 interaction in their tetrameric assembly? How far apart are they from each other? If they are far, the only driving force of linear clustering of TLR3s is dsRNA interactions? If they are at interaction distance, they should validate this interface in cellular assays.

2. The authors mention intracellular and transmembrane part of TLR3 plays no significant role in cluster formation, however, in the discussion they describe the role of BB loop in TIR domain when mutated stops the filament formation and hence, the TLR signaling. This shows the TIR domain interactions are important which may contribute to TLR clustering. Authors should explain how they can rationalize these observations and determine if full length TLR3 clustering is inhibited when TIRs are mutated.

3. Full-length TLR3 was used for structure determination and the authors were not successful in resolving TM or TIR domains. They should explain why they don't see these domains or resolve them by focused refinement.

4. Authors should provide sections of electron density fitted with atomic model.

REVIEWER'S COMMENTS

Reviewer #1:

> **Reviewer's Comment 1.** Since the authors have a cryoEM system setup for full-length TLR3 I recommend (if possible) they also test the effect of antibody and reverse-charge mutants on clustering of full-length TLR3 on double-stranded RNA. This should address whether or not the transmembrane or TIR domain regions also contributes to clustering as suggested by the authors.

(Changes made)

- (1) We conducted a suggested antibody experiment using the full-length TLR3 and added the results to Supplementary Fig. 6. We also added Supplementary Table 5 and Supplementary Fig. 24 for a summary of the data collection.
- (2) We conducted a mutagenesis experiment in the TIR domain and determined the structure of the mutant. The result is added as Supplementary Fig. 10.
- (3) We tried to determine the structures of the NT- and/or CT-mutants for the full-length TLR3. However, the expression levels of the mutants were significantly decreased in our experimental condition and we could not obtain enough cryo-EM samples.
- (4) We added the following paragraphs,

In page 5, lines 122-128.

"A similar result was obtained using full-length TLR3 and a Fab form of mAb12 (Supplementary Fig. 6 and Supplementary Table 5). We also confirmed the previous result, that the mAb12 antibody could inhibit TLR3 activation, in transfected cells, as shown in Fig. 4c and d. This was not due to conformation changes in TLR3 because antibody binding does not disturb the structure or dimerization of TLR3 (Supplementary Fig. 7). Collectively, our structural study and previous reports demonstrate that cluster formation is indispensable for TLR3 signaling."

In page 7, lines 182-200.

"Several factors can contribute to the ordered assembly of TLR3 along the dsRNA strands. First, as shown in Fig. 2, the deletion of both the transmembrane and the intracellular domains did not change the structure of the receptor cluster. This result demonstrates that the ectodomain plays a key role not only in ligand binding and receptor dimerization but also in receptor clustering. The TLR3-RNA interaction alone cannot explain the "ordered" clustering because the entropic effect will randomly distribute the bound TLR3 dimers along the dsRNA strands. The long-range charge interaction between the ectodomains of the TLR3 dimers contributes to clustering as shown in Fig. 5. However, it cannot be the predominant factor under the cellular environment where TLR3 are bound to the membrane as shown by the reporter assays (Supplementary Fig. 9). Second, transmembrane and intracellular domains can play indirect roles. The transmembrane and intracellular domains cannot play direct roles in receptor

clustering because these domains between the adjacent dimeric units are separated more than 100 angstroms apart. However, they can still play an indirect role. For example, the interaction of the intracellular TIR domain of TLR3 with those of the signaling proteins can stabilize the pre-assembled receptor cluster or amplify the size of the cluster by bringing more receptors to the cluster through TRIF homo-oligomerization. Interactions with other cellular components, such as lipid rafts, intracellular cytoskeletons or proteins such as CLEC18A³³ may also have a role in cluster formation and stabilization. More structural and biochemical research is needed on these effects."

In page 6, lines 152-159.

" Previously, a point mutation in the BB loop that connects the helix B and strand B of the TIR domain has been shown to interrupt TLR signaling, including that of TLR3. To investigate the role of this mutation in TLR3 clustering, we produced a BB loop mutant, A795H, of TLR3 and determined the cluster state of the mutant³². As shown in Supplementary Fig. 10 and Supplementary Table 5, the A795H TLR3 formed the ligand-induced cluster with the same spacing and structure as those of the wild-type TLR3. This structure demonstrates that the BB loop mutation is not directly involved in ligand binding or clustering of TLR3 but in its interaction with intracellular signaling proteins."

> **Reviewer's Comment 2.** Many of the conclusions in this study are based on comparing number of particles in clustered and dimeric states from 2D classes analyses of cryoEM data but there is limited method details. Additional details should be provided on the 2D templates used for picking clustered and dimeric particles and how the 2D classification (including numbers of 2D classification rounds) was performed for each dataset.

(Changes made) We added Supplementary Fig. 11 to explain the picking method in detail. We also added the following sentences in Methods.

In page 11, lines 344-358.

"TLR3 particles in the cluster and dimer states were picked using the Topaz particle-picking method³⁸. For this, the TLR3 particles were initially picked by the blob-picking method using the cryoSPARC program. These particles were used to generate the picking models for Topaz. The box sizes for particle picking were adjusted to 360~512 angstroms, which is large enough to accommodate a TLR3 tetramer (Supplementary Tables 1-5). Multiple Topaz iterations were conducted to optimize the accuracy of the initial picking models (Supplementary Fig. 11).

The final picked particles were separated into 50 classes by 2D classification using the cryoSPARC program. Classes of particles with clear and well-centered images were purified by multiple rounds of 2D classifications and refined by the heterogeneous refinement method with multi-class *Ab-initio* models. The TLR3 particles could be categorized into two different states, clustered and dimeric states, by the 2D classification. The accuracy of the state assignment was confirmed by overlaying the coordinates of the picked particles over the representative microscopic images and manually inspecting the clustering states of the particles (Figs. 2-5)."

> **Reviewer's Comment 3.** The authors propose that the linear clustering of ectodomains observed for TLR3 in this study may be a mechanism used by all TLRs (lines 51-53 and 180-191) since they all have intracellular TIR domains. However, only TIR domains of adaptor proteins involved in TLR signalling have been shown to form higher-order assemblies. Such assemblies is yet to be shown for any TLR TIR domains and it is not clear whether or not they need to form larger complexes to induce a robust immune response. Moreover, there is no evidence that other TLR ectodomains cluster upon ligand binding and this seems unlikely for some TLRs that interact with smaller ligands. I recommend the authors moderate/revise these statements.

(Changes made) We agree that the paragraph is somewhat speculative and therefore deleted most of the sentences.

-Before change.

"The sequence conservation of TLR and signaling adaptors suggests that a similar mechanism may work in the activation of other TLR family proteins. Although each member of the TLR family uses a unique structure for ligand interaction, they all use common adaptor proteins, either MyD88 or TRIF, for intracellular signaling³². TLR2 and TLR4 use both for activation. These signaling adaptors and TLRs share a domain called the TIR domain which has high sequence and structural homology^{13,33}. These TIR domains mediate the binding of adaptor proteins to the activated TLRs and oligomerization of the adaptor complexes. A point mutation in the BB loop in these TIR-containing receptors and adaptors disrupted ligand-induced aggregation of the adaptors and stopped TLR signal activation³³⁻³⁶. This suggests that they share a similar structure for the formation of signaling complexes. Based on the sequence homology and previous results, we propose that higher-order clustering of the receptors triggers the activation of not only TLR3 but also other TLR family members."

-After change, in page 8, lines 222-231.

" Two factors may contribute to triggering intracellular signaling. First, as mentioned in the previous paragraph, the clustering of the receptors should enhance the binding avidity of TRIF for the receptors. Second, clustering of the TLR3 TIR domains may increase the productive assembly of intracellular signaling proteins. A previous study showed that TRIF formed a highly ordered and filamentous cluster when activated³⁵. The assembly of the TLR3 cluster with proper spacing may enhance the productive formation of the TRIF filament and trigger signaling. Future structural studies on the activated cluster of receptors and signaling proteins are required for a complete understanding of the immune response mediated not only by TLR3 but also by other TLRs."

> **Reviewer's Comment 4.** Extended data figure 6a-c: Number of particles in total and in the clustered and dimeric states based on the 2D class analyses should be included.

(Changes made) We thank the reviewer's suggestion. We included the numbers of particles in total, clustered, and dimeric states in the micrograph. Please see Fig. 3.

> **Reviewer's Comment 5.** Extended data figures 10-15: These figures should be referenced

in main text when describing the structures. These figures should also include sub-panels showing fit of model to density and map-vs-model fsc curves.

(Changes made) The representative electron density maps and the map-vs-model FSC curves were added. Please see Supplementary Figs. 12-25. These figures were referenced as recommended in the main text. Thank you.

Minor issues:

(1) Lines 63-64. Two values for the final resolution is reported (2.7 or 2.8 Angstrom). Please update

We changed the sentence on page 4, line 63.

(2) Extended data table 3: this table is lacking titles for the three different columns.

We added the column titles in Supplementary Table 3.

(3) Extended data figure 4. In the RNA molecule the colors are too similar. Please update sugar/base colour.

We thank the reviewer's suggestion. As recommended, the sugar/base colors were changed from red to darker red. Thank you.

Reviewer #2:

> **Reviewer's Comment 1.** The displayed structure showed only the ectodomain, without the transmembrane domain and TIR domain. There is structure of full length TLR3 (Ishida H., NSMB, 2021). Maybe the author could consider docking the published structure to acquire a full length TLR3 cluster to observe the interactions between transmembrane domains and TIR domains.

(Changes made) We added the following paragraph in the Discussion, page 7, lines 200-211

"The structure of full-length TLR3 in a complex with UNC93B1 was recently reported by Ishida *et al*³³. UNC93B1 is a chaperone protein involved in the trafficking of TLR3 from the endoplasmic reticulum to the endosome. In the structure, UNC93B1 formed a 1:1 complex with TLR3. The main interaction sites are the transmembrane domain and the luminal linker between the transmembrane and the ectodomains of TLR3. Due to this interaction, the ectodomain of TLR3 made a ~30-degree tilt to the membrane plane. The intracellular TIR domain structure is not visible in the cryo-EM map, presumably due to the structural flexibility in the linker between the transmembrane and the TIR domains. In this structure, TLR3 is forced to stay inactive by binding to UNC93B1. Therefore, it is unlikely that the transmembrane domain of TLR3 maintains a similar structure when released from UNC93B1. In our structure, the transmembrane domain and the connecting linkers are more flexible and were not resolvable in the electron density map."

> **Reviewer's Comment 2.** The author claimed that transmembrane and TIR domains were dispensable for the clustering, however compared from Fig 1a and Fig 2a, it is obvious that full length TLR3 formed much more clusters than did the ectodomain only proteins. So what are the exact functions and interactions of transmembrane domain and TIR domain in clustering?

(Our Opinion) This is an important issue. However, we are trying not to make a "quantitative" comparison between cryo-EM data sets under different buffer conditions. For example, the full-length TLR3 has a detergent in the buffer that can have an effect on the clustering state of the TLR3-dsRNA complex. We are trying to develop a more quantitative assay method for our future research.

(Changes made) We added the following paragraph in the Discussion, page 7, lines 182-198.

"Several factors can contribute to the ordered assembly of TLR3 along the dsRNA strands. First, as shown in Fig. 2, the deletion of both the transmembrane and the intracellular domains did not change the structure of the receptor cluster. This result demonstrates that the ectodomain plays a key role not only in ligand binding and receptor dimerization but also in receptor clustering. The TLR3-RNA interaction alone cannot explain the "ordered" clustering because the entropic effect will randomly distribute the bound TLR3 dimers along the dsRNA strands. The long-range charge interaction between the ectodomains of the TLR3 dimers contributes to clustering as shown in Fig. 5. However, it cannot be the predominant factor under the cellular environment where TLR3 are bound to the membrane as shown by the reporter assays (Supplementary Fig. 9). Second, transmembrane and intracellular domains

can play indirect roles. The transmembrane and intracellular domains cannot play direct roles in receptor clustering because these domains between the adjacent dimeric units are separated more than 100 angstroms apart. However, they can still play an indirect role. For example, the interaction of the intracellular TIR domain of TLR3 with those of the signaling proteins can stabilize the pre-assembled receptor cluster or amplify the size of the cluster by bringing more receptors to the cluster through TRIF homo-oligomerization. Interactions with other cellular components, such as lipid rafts, intracellular cytoskeletons or proteins such as CLEC18A³³ may also have a role in cluster formation and stabilization. More structural and biochemical research is needed on these effects

In page 6, lines 152-159.

" Previously, a point mutation in the BB loop that connects the helix B and strand B of the TIR domain has been shown to interrupt TLR signaling, including that of TLR3. To investigate the role of this mutation in TLR3 clustering, we produced a BB loop mutant, A795H, of TLR3 and determined the cluster state of the mutant³². As shown in Supplementary Fig. 10 and Supplementary Table 5, the A795H TLR3 formed the ligand-induced cluster with the same spacing and structure as those of the wild-type TLR3. This structure demonstrates that the BB loop mutation is not directly involved in ligand binding or clustering of TLR3 but in its interaction with intracellular signaling proteins."

> **Reviewer's Comment 3.** It is showed that the TLR3 dimer structure is quite stable and easy to form. The C-terminal of ectodomain TLR3 and N-terminal of ectodomain interaction account for clustering. Despite TLR3:dsRNA interaction, does TLR3 interact with TLR3 on the opposite side in the dimer unit?

(Changes made) We added the following paragraph in the Discussion, page 7, lines 182-199.

"Several factors can contribute to the ordered assembly of TLR3 along the dsRNA strands. First, as shown in Fig. 2, the deletion of both the transmembrane and the intracellular domains did not change the structure of the receptor cluster. This result demonstrates that the ectodomain plays a key role not only in ligand binding and receptor dimerization but also in receptor clustering. The TLR3-RNA interaction alone cannot explain the "ordered" clustering because the entropic effect will randomly distribute the bound TLR3 dimers along the dsRNA strands. The long-range charge interaction between the ectodomains of the TLR3 dimers contributes to clustering as shown in Fig. 5. However, it cannot be the predominant factor under the cellular environment where TLR3 are bound to the membrane as shown by the reporter assays (Supplementary Fig. 9). Second, transmembrane and intracellular domains can play indirect roles. The transmembrane and intracellular domains cannot play direct roles in receptor clustering because these domains between the adjacent dimeric units are separated more than 100 angstroms apart. However, they can still play an indirect role. For example, the interaction of the intracellular TIR domain of TLR3 with those of the signaling proteins can stabilize the pre-assembled receptor cluster or amplify the size of the cluster by bringing more receptors to the cluster through TRIF homo-oligomerization. Interactions with other cellular components, such as lipid rafts, intracellular cytoskeletons or proteins such as CLEC18A³³ may also have a role in cluster formation and stabilization. More structural and

biochemical research is needed on these effects

> **Reviewer's Comment 4.** The author claimed that cluster boost stronger immune responses, all they did were in vitro experiments. Can the clustering be observed in cells or in physiological conditions?

(Changes made) We added the following paragraph in the Discussion, page 7, lines 170-181.

"Previously, Leonard *et al.* showed that TLR3 had a higher affinities with longer dsRNAs. They measured the binding affinity of dsRNA with lengths of 39, 48, 139, and 540 base pairs at three different pHs²⁷. They found that, at pH 6.0, the binding affinity of 130 bp dsRNA was five times higher than that of 48 bp dsRNA. RNA binding showed strong positive cooperativity, indicating that the affinity increased with the number of TLRs bound. Similarly, Luo *et al.* showed that 139 bp dsRNA but not 49 bp dsRNA induced TLR3-dependent NF- κ B activation²⁹. They showed that two monoclonal antibodies, mAb12 and mAb1068, blocked TLR3 activation without disrupting the binding or dimerization of TLR3. They also showed that mAb12 could block poly(I:C)-induced CCL5 production in NHBE, a bronchial epithelial cell. Jelinek *et al.* showed that dsRNA longer than 90 bp was required to induce cytokine release by dendritic cells and generate antigen-specific cytotoxic T cells in mice²⁸. These results indicate that clustering is critical for the high-affinity binding and robust activation of TLR3."

> **Reviewer's Comment 5.** It seemed the dimer and cluster conformation had no obvious differences, thus how did the protein cluster induce stronger signal activation? Perhaps more downstream molecules?

(Changes made) We added the following paragraph in the Discussion, page 8, lines 222-231.

" Two factors may contribute to triggering intracellular signaling. First, as mentioned in the previous paragraph, the clustering of the receptors should enhance the binding avidity of TRIF for the receptors. Second, clustering of the TLR3 TIR domains may increase the productive assembly of intracellular signaling proteins. A previous study showed that TRIF formed a highly ordered and filamentous cluster when activated³⁵. The assembly of the TLR3 cluster with proper spacing may enhance the productive formation of the TRIF filament and trigger signaling. Future structural studies on the activated cluster of receptors and signaling proteins are required for a complete understanding of the immune response mediated not only by TLR3 but also by other TLRs."

Reviewer #3:

> **Reviewer's Comment 1.** The authors can identify a minimum unit of a TLR3 cluster for triggering the activation of signaling pathways. Is a tetramer formation sufficient, or is the multimer formation (more than hexamers) required? The author can test this by using dsRNAs with different lengths.

(Changes made) We added the following paragraph in the Discussion, page 7, lines 170-181.

"Previously, Leonard *et al.* showed that TLR3 had a higher affinity with longer dsRNAs. They measured the binding affinities of dsRNA with lengths of 39, 48, 139, and 540 base pairs at three different pHs²⁷. They found that, at pH 6.0, the binding affinity of 130 bp dsRNA was five times higher than that of 48 bp dsRNA. RNA binding showed strong positive cooperativity, indicating that the affinity increased with the number of TLRs bound. Similarly, Luo *et al.* showed that 139 bp dsRNA but not 49 bp dsRNA induced TLR3-dependent NF- κ B activation²⁹. They showed that two monoclonal antibodies, mAb12 and mAb1068, blocked TLR3 activation without disrupting the binding or dimerization of TLR3. They also showed that mAb12 could block poly(I:C)-induced CCL5 production in NHBE, a bronchial epithelial cell. Jelinek *et al.* showed that dsRNA longer than 90 bp was required to induce cytokine release by dendritic cells and generate antigen-specific cytotoxic T cells in mice²⁸. These results indicate that clustering is critical for the high-affinity binding and robust activation of TLR3."

> **Reviewer's Comment 2.** An analogue of poly (I:C), poly I:poly C12U, has been recognized as a weak TLR3 ligand. Since the lengths of dsRNA part in poly I:poly C12U is supposed to be shorter than poly (I:C) due to a mismatch, it is interesting to investigate if the TLR3 cluster formation is restricted on this compound.

(Our Opinion) We thank the reviewer's recommendation. It can be a very interesting experiment. However, poly I:poly C12U is a proprietary material and we had a hard time obtaining the research material. We contacted three commercial suppliers (BenchChem, BioTEK, AIM). However, they refused to synthesize it. We also contacted the Ampligen company but could not get their response. We will keep trying to obtain the material for our future research.

> **Reviewer's Comment 3.** The TLR3 signaling leads to the activation of IRF3, which transactivates type I IFN genes and ISGs, in addition to NF- κ B. Thus, the authors need to check if the activation of IFN β promoter is impaired by the inhibition of cluster formation via the antibody treatment and/or mutations. Indeed, the authors described the use of a reporter controlled by the IFN- β promoter in the methods section, although I could not find the data using this reporter in the figures or the main text.

(Changes made) Thank you very much for the comment.

(1) We conducted the requested experiment and added the result as Fig. 4c.

(2) We added the following paragraph to describe the experimental procedure in the Methods, page 12, lines 379-397.

"For the luciferase IFN- β assay, HEK-Blue Null cells (InvivoGen) were seeded at 8×10^4 cells/well in a 24-well plate and co-transfected with 200 ng of TLR3-pcDNA3.1 plasmid, 50 ng of a reporter plasmid containing a firefly luciferase gene under the control of the IFN- β promoter (IFN-Beta_pGL3, Addgene, Plasmid #102597), and 2 ng of Renilla luciferase gene under the control of a constitutive HSV-thymidine kinase promoter (pRL-TK, Promega) using Lipofectamine 2000 according to the manufacturer's protocol (Supplementary Fig. 26). Transfection vectors containing the N-terminal FLAG tag and the human full-length TLR3 genes were cloned into pcDNA 3.1(+) to generate the TLR3-pcDNA3.1 plasmid (Supplementary Table 9). After 24 hours of transfection, the cells were treated with 10 $\mu\text{g}/\text{mL}$ of poly(I:C) (InvivoGen) to stimulate TLR3 signaling. To test the neutralizing effect of the mAb12 antibody, the cells were pretreated for one hour with 5 – 50 $\mu\text{g}/\text{mL}$ of the scFv form of the antibody before poly(I:C) addition. After 24 hours of TLR3 stimulation, the cells were harvested and firefly luciferase and renilla luciferase activity was measured using the Dual-Glo[®] Luciferase Assay System (Promega) according to the manufacturer's protocol. The samples were transferred to a 96-well plate and luminescence signals were measured using a microplate reader (BioTek). Firefly luciferase activity was normalized to Renilla luciferase activity. All data are presented as the mean \pm SEM (error bars). GraphPad Prism 7 (GraphPad Software, USA) was used to perform ANOVA to identify significant differences between groups, as indicated in the figure legends."

(2) To avoid confusion, we also added the following sentences in the Methods, page 12, lines 398-402. The promoter used for the NF- κ B assay contains only the minimal part of the IFN- β promoter and mostly controlled by the NF- κ B binding sites fused to it.

"For the Quanti-Blue NF- κ B assay, HEK-Blue Null cells, which express a secreted embryonic alkaline phosphatase (SEAP) reporter gene under the control of an IFN- β minimal promoter fused to five NF- κ B and AP-1 binding sites, were cultured in Dulbecco's modified Eagle's medium (DMEM; WELGENE) with 5 mM L-glutamate and 1% antibiotic-antimycotic and Zeocin (Thermo Fisher Scientific)."

> **Reviewer's Comment 4.** The authors found that the TLR3 cluster formation is inhibited by mutations in the N-terminal (NT) part as well as in the C-terminal (CT) part. However, the TLR3 CT mutant had no effect on NF- κ B assay as shown in Extended Fig. 9. The authors need to explain the discrepancy between the antibody treatment and the introduction of mutations in the inter-dimer interface. First, the authors should test the effect of NT mutations in NF- κ B activation. Further, the authors can test if these mutations affect the expression of IFN β -reporter.

(Our Opinion) We thank the reviewer's suggestion. We could not obtain enough expression of the NT mutant when transfected to the cell and could not conduct the suggested experiment.

We think the discrepancy between antibody binding and charge-repulsion mutation on TLR3 signaling may originate from a complicated environment of the cell. The charge-repulsion can effectively inhibit the formation of the ectodomains of TLR3 in solution.

However, it may not have enough effect to disrupt cluster formation of the full-length TLR3 anchored on the endosomal membrane. Other than the long-range charge interaction, the following factors can contribute for cluster formation in the cellular environment.

- 1) Restricted diffusion of TLR3 in the heterogeneous and viscous endosomal membrane (Radhakrishnan et al., *Ann. of Biomed. Eng.* 2012:40:2307).
- 2) Interaction with intracellular adaptor proteins – TRIF, TRAF etc.
- 3) Interaction with TLR3 co-receptors or regulatory proteins, for example, CLEC18A (Huang et al., *Commun. Biol.* 2021:4(1):229), MLP (Tang et al., *Circulation.* 2020:141(12):984) and others.

The long-range (6~16 Å) electrostatic interaction between TLR3 dimers are very weak. Compared to this, binding affinity of the antibody is much stronger and cannot be easily disturbed by the endosomal environment.

(Changes made) We added the following paragraph in the Discussion, page 7, lines 182-198.

"Several factors can contribute to the ordered assembly of TLR3 along the dsRNA strands. First, as shown in Fig. 2, the deletion of both the transmembrane and the intracellular domains did not change the structure of the receptor cluster. This result demonstrates that the ectodomain plays a key role not only in ligand binding and receptor dimerization but also in receptor clustering. The TLR3-RNA interaction alone cannot explain the "ordered" clustering because the entropic effect will randomly distribute the bound TLR3 dimers along the dsRNA strands. The long-range charge interaction between the ectodomains of the TLR3 dimers contributes to clustering as shown in Fig. 5. However, it cannot be the predominant factor under the cellular environment where TLR3 are bound to the membrane as shown by the reporter assays (Supplementary Fig. 9). Second, transmembrane and intracellular domains can play indirect roles. The transmembrane and intracellular domains cannot play direct roles in receptor clustering because these domains between the adjacent dimeric units are separated more than 100 angstroms apart. However, they can still play an indirect role. For example, the interaction of the intracellular TIR domain of TLR3 with those of the signaling proteins can stabilize the pre-assembled receptor cluster or amplify the size of the cluster by bringing more receptors to the cluster through TRIF homo-oligomerization. Interactions with other cellular components, such as lipid rafts, intracellular cytoskeletons or proteins such as CLEC18A³³ may also have a role in cluster formation and stabilization. More structural and biochemical research is needed on these effects

> **Reviewer's Comment 5.** Extended Data Fig. 6d shows that the ratio of TLR3 cluster to dimeric state greatly elevated when the concentration of ecto TLR3 increased. The results suggest that there is a threshold in the concentration of TLR3 preferentially forming clusters rather than dimers. Since the authors found that the cluster formation contributes to efficient signaling, it is intriguing to examine if the gradual increase of TLR3 expression in cells leads to the great enhancement of signaling at the certain TLR3 expression level.

(Changes made) We conducted the recommended experiment and added the result as the

Supplementary Fig. 26. We could not find any sign of cooperativity in the TLR3 signalling. We think this is due to complex cellular environment that we cannot reproduce using purified TLR3 and ligands. For example, we cannot control exact amount of active TLR3 produced by the transfection experiment. We also cannot control amount of TLR3 transported to the endosomal membrane, amount of signaling proteins, interactions of TLR3 with signaling proteins, lipid membrane and cytoskeleton *etc.*

Minor comments:

1) TLR3 should be referred to as Toll-like receptor 3 at the first appearance.

Corrected in the page 3, line number 1. Thank you for pointing this out.

2) "r.m.s.d." in line 73 should be spelled out at the first appearance that it stands for root mean square deviation.

Corrected in the page 4, line number 72. Thank you for pointing this out.

3) The scFv on line 119 should be spelled out at first appearance as a single-chain variable domain fragments.

Corrected in the page 5, line number 118. Thank you for pointing this out.

4) The statement on line 147 that "transmembrane sites and intracellular domains are important for cluster formation." is confusing because it is contradictory to the results shown in Fig. 2. In this figure, the authors show that TLR3 lacking these domains still forms clusters. I believe that the clarification of this sentence is required.

We thank pointing out this. We deleted the misleading sentence in the revised manuscript and added the following paragraph in the Discussion, page 7, lines 182-200.

"Several factors can contribute to the ordered assembly of TLR3 along the dsRNA strands. First, as shown in Fig. 2, the deletion of both the transmembrane and the intracellular domains did not change the structure of the receptor cluster. This result demonstrates that the ectodomain plays a key role not only in ligand binding and receptor dimerization but also in receptor clustering. The TLR3-RNA interaction alone cannot explain the "ordered" clustering because the entropic effect will randomly distribute the bound TLR3 dimers along the dsRNA strands. The long-range charge interaction between the ectodomains of the TLR3 dimers contributes to clustering as shown in Fig. 5. However, it cannot be the predominant factor under the cellular environment where TLR3 are bound to the membrane as shown by the reporter assays (Supplementary Fig. 9). Second, transmembrane and intracellular domains can play indirect roles. The transmembrane and intracellular domains cannot play direct roles in receptor clustering because these domains between the adjacent dimeric units are separated

more than 100 angstroms apart. However, they can still play an indirect role. For example, the interaction of the intracellular TIR domain of TLR3 with those of the signaling proteins can stabilize the pre-assembled receptor cluster or amplify the size of the cluster by bringing more receptors to the cluster through TRIF homo-oligomerization. Interactions with other cellular components, such as lipid rafts, intracellular cytoskeletons or proteins such as CLEC18A³³ may also have a role in cluster formation and stabilization. More structural and biochemical research is needed on these effects

We also added the following paragraph in page 6, lines 152-159.

" Previously, a point mutation in the BB loop that connects the helix B and strand B of the TIR domain has been shown to interrupt TLR signaling, including that of TLR3. To investigate the role of this mutation in TLR3 clustering, we produced a BB loop mutant, A795H, of TLR3 and determined the cluster state of the mutant³². As shown in Supplementary Fig. 10 and Supplementary Table 5, the A795H TLR3 formed the ligand-induced cluster with the same spacing and structure as those of the wild-type TLR3. This structure demonstrates that the BB loop mutation is not directly involved in ligand binding or clustering of TLR3 but in its interaction with intracellular signaling proteins."

Reviewer #4:

> **Reviewer's Comment 1.** Did author analyze the dimer1-dimer2 interaction in their tetrameric assembly? How far apart are they from each other? If they are far, the only driving force of linear clustering of TLR3s is dsRNA interactions? If they are at interaction distance, they should validate this interface in cellular assays.

(Changes made)

(1) We added the following sentences in the Discussion, page 7, lines 182-200.

"Several factors can contribute to the ordered assembly of TLR3 along the dsRNA strands. First, as shown in Fig. 2, the deletion of both the transmembrane and the intracellular domains did not change the structure of the receptor cluster. This result demonstrates that the ectodomain plays a key role not only in ligand binding and receptor dimerization but also in receptor clustering. The TLR3-RNA interaction alone cannot explain the "ordered" clustering because the entropic effect will randomly distribute the bound TLR3 dimers along the dsRNA strands. The long-range charge interaction between the ectodomains of the TLR3 dimers contributes to clustering as shown in Fig. 5. However, it cannot be the predominant factor under the cellular environment where TLR3 are bound to the membrane as shown by the reporter assays (Supplementary Fig. 9). Second, transmembrane and intracellular domains can play indirect roles. The transmembrane and intracellular domains cannot play direct roles in receptor clustering because these domains between the adjacent dimeric units are separated more than 100 angstroms apart. However, they can still play an indirect role. For example, the interaction of the intracellular TIR domain of TLR3 with those of the signaling proteins can stabilize the pre-assembled receptor cluster or amplify the size of the cluster by bringing more receptors to the cluster through TRIF homo-oligomerization. Interactions with other cellular components, such as lipid rafts, intracellular cytoskeletons or proteins such as CLEC18A³³ may also have a role in cluster formation and stabilization. More structural and biochemical research is needed on these effects."

(2) We added the following sentences on page 6, lines 152-159.

" Previously, a point mutation in the BB loop that connects the helix B and strand B of the TIR domain has been shown to interrupt TLR signaling, including that of TLR3. To investigate the role of this mutation in TLR3 clustering, we produced a BB loop mutant, A795H, of TLR3 and determined the cluster state of the mutant³². As shown in Supplementary Fig. 10 and Supplementary Table 5, the A795H TLR3 formed the ligand-induced cluster with the same spacing and structure as those of the wild-type TLR3. This structure demonstrates that the BB loop mutation is not directly involved in ligand binding or clustering of TLR3 but in its interaction with intracellular signaling proteins."

(3) We added the following sentences on page 6, lines 133-135.

"The distance between these charged areas is more than 6 angstroms and appears too far to be considered a strong interaction. However, previous studies showed that even these weak

electrostatic forces can modulate protein-protein complex formation^{30, 31}.

> **Reviewer's Comment 2.** The authors mention intracellular and transmembrane part of TLR3 plays no significant role in cluster formation, however, in the discussion they describe the role of BB loop in TIR domain when mutated stops the filament formation and hence, the TLR signaling. This shows the TIR domain interactions are important which may contribute to TLR clustering. Authors should explain how they can rationalize these observations and determine if full length TLR3 clustering is inhibited when TIRs are mutated.

(Changes made) We added the following sentences in the Results, page 6, lines 152-159.

" Previously, a point mutation in the BB loop that connects the helix B and strand B of the TIR domain has been shown to interrupt TLR signaling, including that of TLR3. To investigate the role of this mutation in TLR3 clustering, we produced a BB loop mutant, A795H, of TLR3 and determined the cluster state of the mutant³². As shown in Supplementary Fig. 10 and Supplementary Table 5, the A795H TLR3 formed the ligand-induced cluster with the same spacing and structure as those of the wild-type TLR3. This structure demonstrates that the BB loop mutation is not directly involved in ligand binding or clustering of TLR3 but in its interaction with intracellular signaling proteins."

> **Reviewer's Comment 3.** Full-length TLR3 was used for structure determination and the authors were not successful in resolving TM or TIR domains. They should explain why they don't see these domains or resolve them by focused refinement.

(Our Opinion) We tried to reveal the TM and TIR domain structures using focused refinement, local refinement, and homologous refinement with a mask using cryoSPARC. However, we could not obtain reliable density for the TM and the TIR domains. There are several possible reasons we could not obtain TM and TIR domain structures:

- (1) Flexibility between ectodomain and TM domain
- (2) Flexibility between TM and TIR domain
- (3) Low signal-to-noise ratio of TM in the detergent micelle

Current cryo-EM structure studies for other membrane receptors without binding partners reveal only the extracellular domain as well. For example, recent structures of Insulin receptor (Uchikawa et al., 2019), IGF-1 Receptor (Zhang et al., 2020), and Epidermal growth factor receptor (Huang et al., 2021) showed only the extracellular domains. The cryo-EM structure of TLR3 in a complex with a chaperone protein, UNC93B1, (Ishida et al., 2021) revealed the structure of the transmembrane domain. This is mainly due to UNC93B1 that interacts with the TM domain of TLR3 and stabilizes its structure. Even in this case, they failed to resolve the intracellular TIR domain in the electron density map.

(Changes made) We added the following sentences in the Discussion, page 7, lines 201-212.

"The structure of full-length TLR3 in a complex with UNC93B1 was recently reported by

Ishida *et al*³⁴. UNC93B1 is a chaperone protein involved in the trafficking of TLR3 from the endoplasmic reticulum to the endosome. In the structure, UNC93B1 formed a 1:1 complex with TLR3. The main interaction sites were the transmembrane domain and the luminal linker between the transmembrane and the ectodomains of TLR3. Due to this interaction, the ectodomain of TLR3 made a ~30-degree tilt to the membrane plane. The intracellular TIR domain structure was not visible in the cryo-EM map, presumably due to the structural flexibility in the linker between the transmembrane and the TIR domains. In this structure, TLR3 was forced to stay inactive by binding to UNC93B1. Therefore, it is unlikely that the transmembrane domain of TLR3 maintains a similar structure when released from UNC93B1. In our structure, the transmembrane domain and the connecting linkers are more flexible and were not resolvable in the density map. "

> **Reviewer's Comment 4.** Authors should provide sections of electron density fitted with atomic model.

(Changes made)

The representative electron density superimposed with refined atomic models were added. Please see Supplementary Figs. 12-23.

REVIEWERS' COMMENTS

Reviewer #1 (Remarks to the Author):

The manuscript is improved and I am happy to recommend acceptance subject to the following minor change:

Supplementary figures 12-25 should be referenced in the main text when the structures are first described.

Reviewer #2 (Remarks to the Author):

The authors explained the reasons why they did not obtain the TM and TIR domain structures of TLR3 due to technical difficulties.

The authors also modified their statements on the factors influencing TLR3 clustering, not only the ectodomain-RNA interaction and the mutual interaction of ectodomains. They also did extra experiments to obtain the structure and clustering status of the A795H mutant, revealing that the BB loop mutation was not directly involved in ligand binding or clustering of TLR3.

They cited several papers to support that TLR3 clustering induced by longer dsRNAs induced stronger activation of signaling.

Overall, I feel that the authors have adequately addressed most of my concerns.

Reviewer #3 (Remarks to the Author):

The revised manuscript is substantially improved. I think this manuscript is ready for publication in Nature Communications.

Reviewer #4 (Remarks to the Author):

Kudos to all the authors of this fantastic study. All of my concerns are addressed.